# Are patients ready for discharge from the hospital after fast-track total knee arthroplasty?-A qualitative study

Simeng You[1], Na Li[2], Manjie Guo[3], Hong Ji [2,4]*

1 The affiliated hospital of Jiaxing University, The First Hospital of Jiaxing, Jiaxing, Zhejiang, China,
2 Nursing department, The First Affiliated Hospital of Shandong First Medical University, Jinan, Shandong, China, 3 Beijing Hospital of Traditional Chinese Medicine, Capital Medical University, Beijing, China,
4 School of Nursing and Rehabilitation, Shandong University, Jinan, Shandong, China

* honghongji-2005@163.com

**Data Availability Statement:** This qualitative study was conducted in the orthopedic outpatient clinic of a general hospital in China, and because the data contained potentially identifiable or sensitive patient information, participants did not consent to have

## Abstract

### Background

The fast-track based on evidence-based medicine, has dramatically reduced the length of stay for patients undergoing total knee arthroplasty (TKA). Therefore, patients must assume the responsibility for self-functional exercise and care as early as possible. Also, higher standards and expectations of care delivery have been set. Studies into patients' experiences when faced with a discharge decision under a fast-track program are lacking.

### Objectives

(1) Increase the knowledge about patients' experiences of discharged from hospital via a fast-track process after TKA. (2) Explore what gaps exist in the current discharge preparation care service for TKA under fast-track and what can be improved.

### Methods

A qualitative research design was chosen to conduct semi-structured face-to-face interviews with 21 patients from one Chinese hospital who successfully underwent TKA and received discharge orders. Interview data were meticulously analyzed, summarized and thematically distilled using Interpretative Phenomenological Analysis (IPA).

### Results

Three themes emerged from the structural analyses: a) Preparing for discharge despite concerns about symptoms-a sense of joy at discharge despite feelings of helplessness, stigmatisation, anxiety about prosthetic function. b) Managing the rehabilitation difficulties-vigilance is needed for medication management, environmental changes, and intimate relationships. c) Creating conditions for safe transition-compassionate bedside manner, listening to patients, and providing a humanized continuing care and referral services are important for safe transitions.

their full transcripts made public or shared with anyone outside the research team. Therefore the data could not be shared publicly. Data are available from the School of Nursing and Rehabilitation at Shandong University Ethics Committee (contact via Email: m13701533540@163.com, Phone: 86-731-88382000) for researchers who meet the criteria for access to confidential data.

**Funding:** This study is funded by the Medical and Health Science and Technology Development Program of Shandong Province, China (202014021227). The funders had no role in study design, data collection and analysis, decision to publish, or preparation of the manuscript.

**Competing interests:** The authors have declared that no competing interests exist.

## Conclusion

Findings suggest that patients undergoing fast-track TKA report good discharge preparation experiences. However, closer analysis reveals difficulties with this process and important directions in which discharge readiness care services can strive.

## Introduction

TKA is considered the most effective treatment for severe end-stage knee osteoarthritis, addressing severe pain, impaired joint function, significant deformities, and instability resulting from joint diseases [1, 2]. Using available 2000–2014 NIS figures, the U.S. projects that primary TKA is projected to grow 85%, to 1.26 million procedures, by 2030 [3, 4]. In China, there has also seen a rapid rise in TKA surgeries, escalating from 50,000 cases annually a decade ago to nearly 400,000 cases [5].

"Fast-track" [6] refers to a well-established, safe perioperative care strategy involving multi-disciplinary collaboration. Since its introduction in the late 1990s, it has gained widespread acceptance for its ability to achieve several key benefits, including shorter hospital stays [7, 8], reduced mortality [9], and fewer postoperative complications [10]. It is now recognized as a standard of care in orthopedics [11]. Consequently, with the increasing number of TKAs in China, the concept of fast track surgery (FTS) has been widely adopted in this field.

While reaping the clinical benefits of this approach, it's essential to remain vigilant about the challenges associated with centralized clinical pathways. Firstly, shorter hospital stays for TKA patients do not equate to reduced perioperative care but instead require significant organizational and administrative efforts by healthcare staff involved in patient discharge [12]. Additionally, patients are also expected to assume responsibility for engaging in functional exercises and self-care as early as possible, with their readiness for discharge being critical to ensure the continuity of subsequent rehabilitation. Previous studies consistently highlight that patients frequently don't feel adequately prepared for discharge [13], with limited attention to patients' subjective experiences [14]. Rushed discharges have been linked to elevated readmission rates [15], increased postoperative complications, and unexpected healthcare resource utilization [16]. Weiss [15] and Coffey [17] have advocated that incorporating patients' own perspectives in evaluating discharge readiness reduces unplanned readmission and mortality rates, as well as optimizing the discharge planning process while improving patient satisfaction. Recognizing that information is closely tied to trust, it is essential to prioritize listening to and understanding the signals conveyed by patients.

Current studies have predominantly focused on patients' experiences related to pain and rehabilitation after short-term discharge, there appears to be limited understanding patients' experience and needs at the time of receive discharge instructions. It is reasonable to assume that the stress and concerns expressed by patients when facing the decision to be discharged will be more reflective of the current care challenges. This study aims to describe patients' experiences of discharged from hospital following TKA under FTS in China. Additionally, this study attempts to explore what gaps exist in current discharge preparation care service and what could be improved.

## Methods

### Design

This study is a qualitative research methodology that employs Interpretative Phenomenological Analysis (IPA), which aims to gain a deeper understanding of individual experiences and

subjective perceptions. This method is typically used to explore how individuals interpret and ascribe meaning to events, emotions, and phenomena they encounter [18]. Furthermore, it has gained prominence and widespread use over the past decade to fifteen years [19]. The focus of this study is to delve into the subjective construct of the discharge experience as perceived by each participant, which encompasses recollections, impressions, sensations, and fragmentary statements of others. This aligns well with the strengths and capabilities of IPA.

To ensure rigorous and transparent reporting of the research process, the study adhered to the Consolidated Criteria for Reporting Qualitative Research (COREQ) checklist [20] (S1 Checklist).

## Participants

We purposively recruited patients who underwent fast-track TKA between May 2023 and July 2023 in the orthopaedic wards of a tertiary A-level hospital located in Jinan, Shandong Province, China. Patients with ASA classification of class I, II and stable III were enrolled in the fast-track program (S1 File). The hospital is a provincial general hospital open to the public with a specialist orthopaedic inpatient unit and a systematic, standardized perioperative care process specializing in the treatment of joint replacements, joint revisions and wound infections. Patients of different ages, educational levels, and living situations were selected to ensure a diverse sample. The inclusion criteria were as follows: a) receive primary unilateral TKA for advanced KOA; b) under the fast-track program; c) receipt of a discharge order; d) demonstrate effective communication skills; and e) willingness to join the study and consent in writing.

The sample size was determined by considering the participants' ability to provide relevant information, ensuring that each participant's data met the study's requirements [21]. Furthermore, Morse et al. [22] demonstrated that when using a semi-structured interview approach, each participant tends to provide a more constrained and limited account of their experiences compared to an open-ended interview. As a result, in this study, a larger number of interviews was deemed necessary to achieve data saturation. Data saturation is reached when no new themes or crucial information emerge from the interviews.

## Procedure

In this study, all interviews were conducted by a female author (SMY), a third year student for a Master's degree in Nursing at University. Prior work experience of six months in an orthopaedic internship. Received extensive theoretical assessment on qualitative research from the School of Nursing and systematic training on interviewing skills from the First Affiliated Hospital of Shandong First Medical University. None of the participants were known to the interviewer.

Patients were warmly invited to participate in interviews when the researcher (NL) received and verbally explained the discharge instructions (S2 File). Patients were provided with comprehensive information about the study's purpose, significance, and process, and were invited to participate if they expressed interest. Interviews were conducted within 24 hours of receiving the physician's discharge order. To establish a connection with the participants and encourage them to openly express their feelings and thoughts, the interview began with general questions. Throughout the process, the interviewer (SMY) actively suspended preconceived thoughts or experiences, adhering to the principle of attentive listening. Face-to-face interviews were conducted in a relatively comfortable and quiet departmental conference room, typically involving one participant at a time and lasting approximately 30–45 minutes. To ensure construct validity in the qualitative interviews, a pilot study was conducted on three

patients. The final interview guide was co-developed by all the authors based on the research questions, and careful consideration was given to all issues in this area, which included the patient's surgical situation and outcome assessment, experience and assessment of discharge, and consideration of the patient's support level and dilemmas in terms of caregivers, family members, and social support, etc. To minimize the likelihood of missing any potentially relevant factors (e.g., pain management, surgical wound treatment, respiratory therapy, and proper medication were listed as medical factors affecting the later stages of discharge readiness) were pre-included to elicit more thought and response during the pilot study. The "yes or no" questions were further designed with open-ended questions (e.g., Do you feel ready for discharge? If not, what do you think needs to be done better?) The complete interview guide is available in S3 File.

## Data collection and processing

Data collection and processing were carried out by two researchers (SMY and NL). Data were gathered using various tools, such as mobile phone audio, pen and paper notes during the interview process. Additionally, meticulous attention was paid to observing and documenting the patients' voice tones, expressions and body movements.

Within 24 hours of conducting the interviews, two researchers collaborated in transcribing, coding and analyzing the recorded content. The data were transferred to NVivo11 software for this purpose. The primary author (SMY) carefully read through each text multiple times to develop a comprehensive understanding of the content. Subsequently, meaningful statements within the text were highlighted to capture the essential elements. These significant elements were then condensed into clear and concise units of meaning while considering their contextual connections. Units with similar meanings and recurring instances formed sub-themes. Finally, the underlying meanings within the sub-themes were contemplated, leading to the identification of phenomena that could serve as relevant headings, which were then unified as themes [23, 24]. The second author (NL) was engaged in the process from its inception, including the design phase of the research questions and interview guide. Throughout this process, the main responsibility was to review notes and provide comments on the main findings, and if necessary, follow-up interviews as needed to verify the results. Consistency in the results was achieved through the collaboration of both researchers and the comprehensive description of the data, thereby confirming the transfer ability of the findings.

## Ethical considerations

The Ethics Committee of the School of Nursing and Rehabilitation at Shandong University approved this study (No. 2023-R-021). Rigorous adherence to ethical principles was maintained throughout the study. Informed consent was diligently obtained from all participants, with researchers providing a clear and unconditional explanation of the study's objectives. A solemn commitment to confidentiality was made, and participants were informed of their right to withdraw from the study at any time without any consequences. To protect the privacy of the patients, their names were anonymized with strict measures in place.

## Results

### Characteristics of participants

The final analysis included all 21 participants, with no dropouts. Their ages ranged from 53 to 80 years, and the majority of patients were female, living only with their spouses, and were usually interviewed on the postoperative day 3. Table 1 presents the self-reported demographics

**Table 1. Demographic and characteristics of the participants (N = 21).**

| Participants | | n = 21 |
|---|---|---|
| Sex | Male | 8 |
| | Female | 13 |
| Age, years | Median | 68 |
| | Range | 53–80 |
| Educational levels | Elementary school and below | 11 |
| | Junior high school | 6 |
| | High school (including junior college) | 2 |
| | University (including college) and above | 2 |
| Marital status | Married | 17 |
| | Widowed | 4 |
| Living arrangements | Only with spouse | 12 |
| | With family members | 5 |
| | Alone | 2 |
| | With live-in care worker | 2 |
| Residence area | City | 12 |
| | Rural | 9 |
| Hospital length of stay, days | Median | 3 |
| | Range | 2–5 |
| Number of chronic diseases | 0 | 6 |
| | 1~2 | 9 |
| | 3 | 6 |

and characteristics of the participants. Data saturation was achieved after conducting 21 interviews, with no new data emerging thereafter. No repeat interviews were conducted.

## Overview of themes

The interview process naturally provided a comprehensive understanding of patients' experiences of preparing for discharge, three themes emerged:1) Preparing for discharge despite concerns about symptoms, 2) Managing the rehabilitation difficulties, and 3) Creating conditions for safe transition. The S1 Table illustrates how these themes emerged.

**Preparing for discharge despite concerns about symptoms.** *Satisfaction with improved functioning and a sense of reborn joy*. In the interview reports, patients expressed satisfaction with the improvement in knee function.

"I feel like I'm healing day by day. . .(The third day after the surgery)" (ZC, Male, 68 years old, Post-op day 4).

"my surgeon told me that my test results were good, and rehab exercises were on track. . ." (GWX, Male, 57 years old, Post-op day 3)

Patients tend to compare themselves to fellow patients or sense the beginnings of an impending discharge from conversations with nurses or rehabilitation physicians when making their self-assessment.

"The guy next to me had his surgery a day before me, and he got sent home yesterday. . .I guess it's probably my turn to go home." (ZYJ, Female, 63 years, live only with spouse, Post-op day 3)

"The physician said that I can bend (90˚) and straighten (180˚) my knee now, I would be out of the hospital soon." (LW, Male, 69 years old, Post-op day 4)

As the topic of discharge came up, they felt a sense of reborn joy. Expressions such as "going home comes first" and "looking forward to being discharged" were frequently used.

"The surgery went well. . . Getting home is what I'm really looking forward to." (ZTA, Male, 66 years, live only with spouse, Post-op day 3)

"I'm all set to head home now, going home comes first." (ZYX, Female, 54 years, live only with spouse, Post-op day 2)

*Feeling of helplessness*. The influx of information or heightened sensitivity to physical symptoms often generated a sense of vigilance and uncertainty among participants who were overwhelmed.

"It happened so quickly. . . If only they had let me stay one more day, I think I could have absorbed all this info. . ." (SXQ, Female, 69 years, live alone, Post-op day 3)

"I'm uncertain about. . . cause the skin on this leg is numb and doesn't quite feel like mine, but maybe that's just part of the deal." (WTY, Female, 61 years, live with family members, Post-op day 2)

Participants also highlighted their feelings of helplessness in the face of future self-care.

"I'm not so sure I would have done subsequent rehab movements on my own. A sense of being left to its own devices." (DJ, Female, 71 years old, Post-op day 3)

*Stigmatisation*. Participants also tend to associate these physical body image changes with daily socialisation after discharge, which can be potential stigmatising for them.

"They might think I'm in the same boat as someone with a disability." (ZYX, Female, 54 years, live only with spouse, Post-op day 2)

In turn, they argued that the majority of people on the ward had undergone joint surgery and that there were similarities between them, therefore they would not be viewed differently.

"In the hospital, we're all in the same boat, and there's no judgment like that (discriminated against)". (CBC, Female, 78 years old, Post-op day 3)

*Anxiety about prosthetic function*. As most of the participants lacked a medically relevant social background and were undergoing joint replacement surgery for the first time, they voiced anxiety regarding the functionality and durability of their prostheses.

"Can I really go for a deep squat? Could it possibly mess up this fake knee joint? I'm not entirely convinced this artificial joint is as great as the doc claims it to be." (DJ, Female, 71 years, live only with spouse, Post-op day 3)

"I'm kinda scared that if I push it too hard, I might end up needing another replacement. . ." (ZYJ, Female, 63 years old, Post-op day 3)

**Managing the rehabilitation difficulties.** *Highly motivated and currently on the road to recovery*. Postoperative rehabilitation requires continuous and incrementally challenging training. Patients need to possess exceptional motivation and discipline as essential qualities. Interviews revealed that patients were spurred to enact constructive changes by their experiences of traumatic growth. They expressed a keen willingness to adhere to self-management recommendations and exhibited a strong drive to persevere in their rehabilitation regimen, thus embarking on a path to recovery.

"I'm feeling great and totally ready to keep up with the exercises, even if it means dealing with some intense pain while at home this time." (YMY, Female, 65 years, live only with spouse, Post-op day 2)

"This surgery is a big deal in my life, I'm absolutely committed to my post-surgery rehab. . ." (GXP, Female, 68 years, live only with spouse, Post-op day 3)

"Post-op rehab is a big deal, and I can't afford to lose in this battle."(YM, Male, 74 years old, Post-op day 3)

*Medication management*. Nevertheless, various unsettling and disruptive factors can pose significant barriers. These include a biased comprehension of medication dosages, such as excessive caution or relying on painkillers.

"I barely noticed the minor pains, and as long as it's not particularly painful, I'll opt out of pain medication because painkillers are a chemical." (ZWQ, Female,73 years, live alone, Post-op day 4)

"I'll admit, I was feeling scared. Requested my doctor to give me an extra dose of painkillers. . ."(LYQ, Male, 56 years old, Post-op day 2)

Actually, most patients do not realise that the pain can be intense, especially at night.

"Well rested is important, (The first night after surgery) I didn't really get into a deep sleep and the pain got worse, it's quite a struggle." (XZZ, Female, 68 years, live with other family members, Post-op day 3)

*Environmental changes*. The hospital ward environment can be advantageous for patients, and the insufficient adaptation of their homes for post-surgery recovery frequently leads to delayed recuperation following hospital discharge. Ensuring a safe environment for conducting activities was a significant consideration for the participants.

"I mean. . . at home there isn't a strict no-smoking policy like there is in hospitals, where everything around you has handles and is sterilised regularly." (DJ, Female, 71 years, live only with spouse, Post-op day 3)

"That walker won't fit in our bathroom at home, and I'm worried I might take a tumble without it." (JH, Female, 53 years, live only with spouse, Post-op day 2)

The change in surroundings also signifies a shift in dietary patterns and lifestyle habits, and participants also raised concerns related to home-cooked meals.

"My doc and nurse told me to stick to a high-protein and light diet. But you know, I'm a fan of spicy food." (XLE, Male, 62 years, live only with spouse, Post-op day 3)

*Strained social and intimate relationships*. The majority of participants reported a significant reduction in socializing with friends and family relatives due to limitations in mobility and feelings of frustration.

"I won't be attending for too many gatherings because I hardly go out on my own anymore." (YS, Male, 67 years, live only with spouse, Post-op day 3)

An "ideal spousal model of recovery" resulted in spouses often taking on the important role of providing informal care free of charge, and participants mentioned that they suspended some physical intimacy to recover from the illness, but that did have an impact on the intimacy of the couple.

"My husband. . . tried not to share the same bed with me. . . he was worried about accidentally hurting my wounds during the night. . . Sometimes, it does get a bit lonely." (XGB, Female, 69 years, live only with spouse, Post-op day 3)

Moreover, participants exhibited greater sensitivity to the impact of their illness and were often conscious of the concessions made by their family members to provide care, including time off work, financial support, and physical assistance, which engendered a sense of indebtedness.

"My son burned through all his saved-up vacation days. . . he's given up his precious time and spent a lot of money." (LW, Male, 69 years, live with other family members, Post-op day 4)

"my daughter was busy with work. . .but she put a lot of effort into this." (LYQ, Male, 56 years old, Post-op day 2)

**Creating conditions for safe transition.**   *The caring bedside manner of the nursing staff.* Participants generally perceived the management information provided by individuals with expert knowledge, such as attending physicians, rehabilitation specialists, and nurses, as reliable. Participants' perceptions of their illnesses were largely derived from explanations given by this group, and participants indicated that it was the attentive bedside manner of caregivers that created predictability and confidence in the patient's rapid discharge from the hospital.

"They actually came right to my bedside, kept repeating instructions in case I forgot, treating me like one of their own family. . ." (GYH, Female, 80 years, live with live-in care worker, Post-op day 5)

"The nurses taught me how to move, demonstrated with their own bodies at my bedside and constantly shared success stories of other patients. . . that really boosted my confidence." (GWX, Male, 57 years, live only with spouse, Post-op day 3)

**Listening to patients.**   In this study, participants emphasized the significance of the care team listening to their input and desires, as evidenced by communicating the timing of medication changes and choosing more appropriate means of communicating health information.

"I didn't want to start rehab before the med change because I knew it would make the pain worse. . . The nurse understood my concern and shifted the schedule." (XGB, Female, 69 years, live only with spouse, Post-op day 3)

"The nurses handed us a bunch of booklets, but I've got presbyopia. So, they hooked me up with a bunch of rehab videos. . ." (ZTA, Male, 66 years, live only with spouse, Post-op day 3)

Participants reported that when the care team learnt about their needs, they were given special consideration and felt valued.

"The benches in the ward were too low for me, probably because of my height (187 cm). . .I mentioned this to the nurse, and she somehow managed to find me a taller chair." (XLE, Male, 62 years old, Post-op day 3)

*A humanised continuing care and referral services*. Backed by a humanised continuing care and referral services. Patients attempt to establish communication channels with liaison officers to address their concerns after discharge.

"I was wondering if I could have their contact info because they know my health situation better than anyone else." (SXQ, Female, 69 years, live alone, Post-op day 3)

In addition, participants indicated that they were informed of the rehabilitation progress to be followed up by outpatient review and telephone follow-ups, which will contribute to achieve a seamless transition.

"Luckily, the doctor created a WeChat group for us in advance. we can see the time when the attending doctor sits, and go to the outpatient clinic for review when the time for review arrives." (LYQ, male, 56 years, live only with spouse, Post-op day 2)

## Discussion

This study presents the discharge readiness experience of patients undergoing fast-track TKA. It adds valuable knowledge about the understanding of how patients treated with TKA under fast track setting when facing discharge, in addition to revealing the difficulties patients face, it also adds some positive responses about clinical care practices.

The majority of participants reported favorable postoperative recovery experiences and expressed a readiness to be discharged from the hospital. This is similar to the findings of previous studies [25, 26] where the majority of patients reported that it was good to go home and take responsibility for their recovery. There are two plausible explanations for these positive results. First, it is possible that the perioperative multidimensional education program and the multidisciplinary collaborative management approach implemented within the framework of the fast-track concept facilitated patients in tapping into their potential to manage their conditions and acquire skills during the recovery phase [25]. Second, besides the cost of restorative surgery, bed occupancy, nursing services, and medical treatment during postoperative hospitalization place a similar financial burden on the patient. Most cost-effectiveness studies [27, 28] have used data on length of stay to calculate costs. The length of a patient's postoperative hospital stay is considered a good indicator of healthcare costs, and the prospect of early

discharge as a cost-effective rehabilitation option may have motivated them to prepare for discharge. Significantly, behind their positive response to the prospect of discharge, it became evident that various psychological issues were at play, including feeling of helplessness due to uncertainty about the medical condition, fear of social stigma, and heightened concerns about knee prosthesis. The same paradox was found in the study by Woolhead [29], where individuals reported their outcome from TKR as good, but further discussion revealed concern and discomfort with continuing pain and mobility difficulties. Such findings highlight a number of important issues related to the evaluation of discharge readiness. A mixed study [30] conducted in the Malaysian exploring the needs of orthopedic patients at discharge, found that there were emotional, physical and spiritual preparation needs. What's more, in a study of nurses' and patients' perceptions of the discharge readiness, respectively, it was found that nurses rated discharge readiness higher than patients did [31]. These revelations prompted a reconsideration of the healthcare model in place, raising questions about whether a genuinely "patient-centered" [32] approach was being followed and whether healthcare professionals might have overestimated patients' readiness for discharge while potentially overlooking their underlying mental health issues. In light of these findings, we offer several recommendations. Firstly, we recommend that healthcare professionals prioritize open dialogue with patients and remain attentive to the emotional fluctuations within the patients' inner world. Secondly, it is essential to provide more comprehensive and detailed guidance to enhance patients' confidence in coping with feelings of helplessness, stigma, and anxiety about prosthetic function. Additionally, we suggest that, during the postoperative hospitalization period, the flow of information should be managed thoughtfully to prevent overwhelming patients with excessive information.

Our findings also imply that patients' readiness for discharge from hospital is not solely orientated by the caring behaviors of individual healthcare professionals. It is also influenced by the patients' subjective will, pain management, changes in their surroundings, and support from family and friends. In this study, we were pleased to hear that patients were strongly motivated to undertake rehabilitation activities, such as "willingness to persevere without compromise". However, this strong intrinsic motivation may have been somewhat countered by patients' resistance to use analgesics and the challenges posed by the external environment and social relationships. For example, some patients didn't take pain management seriously and resisted using analgesics, despite repeated messages conveyed to them that pain should be nipped in the bud. Additionally, a study similarly revealed that with regard to medication management, some patients held varying interpretations of medication adherence or considered alternative therapies, although they expressed intentions to follow prescribed medication schedules [33]. In this regard, it has been pointed out that optimization and reduction of pain is prerequisite for further development of TKA with rapid rehabilitation [34]. These findings caution that discharge planning teams should comprehensively consider the patient's social context, assess their health literacy regarding pain management, efficiently coordinate analgesic prescriptions and medication regimens, and provide tailored education on medication use [35]. Furthermore, it's crucial to consider the patient's surrounding environment and family support. Our analysis revealed a recurring pattern of inadequate support, responding to gradual tension in family relationships and challenges in adapting to an uncertain living environment, aligning with findings by McKeown [36]. When examining the underlying causes, it became apparent that elderly participants, characterized by vulnerability and reduced mobility, often assumed a passive role in healthcare activities, faced difficulties in comprehending medical information, and frequently sought reassurance and assistance from external sources. These findings underscore the significance of the family's role as a valuable resource in ensuring post-discharge care [30, 37]. Therefore, we recommend that healthcare professionals

actively involve the family in developing home rehabilitation plans and conveying self-management instructions [38]. Additionally, it's crucial to assess whether patients have access to external support, such as environmental modifications that promote safe activities [39].

Through a review and analysis, we've identified a "know-do gap" whereby our resident clinicians and nurses sometimes struggle to execute tasks despite knowing what should be done [40]. This "know-do gap" is also observed in our cultural context, where the final decision to discharge may be sudden, and standardized discharge care processes are challenging to implement fully due to understaffing in our hospital beds [41]. However, from the interviews conducted for this study, it appears that the current nursing teams effectively utilize the hospitalization period. They exhibit valuable qualities such as bedside manner, attentive listening to patients' voice, and a humanized continuing care and referral services, which help bridge this gap. Increasingly, academia is recognizing the importance of valuing patients' experiences and preferences in delivering high-quality care [42–44]. Attentive bedside manner and listening to patients allows healthcare providers to respond promptly to their urgent needs [45]. This leads us to the idea that empathy and the ability to think differently in healthcare professionals can be learned and impacted as a teachable communication skill rather than a personality trait. Finally, in order to better help patients achieve a smooth transition, we have to consider the patients' needs for continuity of care and referral services. Encourage medical institutions to give full play to their advantages in professional technology and talents, provide patients with humane and diversified nursing services, gradually improve the content and mode of services, including but not limited to WeChat-based instant information exchange platform to ensure the continuity of care services.

## Strengths and limitations of the study

This study has several limitations that should be acknowledged. Firstly, the participants were recruited exclusively from one hospital, and the results are limited to the Chinese health care system. This limitation might raise concerns about the generalizability and broader significance of the findings. However, by selecting patients of different ages, educational levels, and living conditions, as well as pre-surveying prior to the commencement of the formal interviews to try to safeguard the transferability of the findings. Secondly, while this study aimed to identify deficiencies in the discharge services provided by the care team and develop relevant recommendations, it did not include interviews with care team members, particularly care managers. Relying solely on the patients' perspective might restrict the applicability of the recommendations, but the results of this study are still instructive. In future research, it would be valuable to combine experiences of patients with those of healthcare professionals to gain a comprehensive understanding of priorities for service improvement. Lastly, despite interviewers clearly explaining to participants that they were not responsible for physical health assessments within the unit, participants may have still reported what they thought the researchers wanted to hear. This potential bias should be considered when interpreting the study's findings.

In addition, this study possesses notable strengths. Firstly, the interview questions were intentionally designed to be open-ended, allowing participants the freedom to express their discharge preparation experiences in greater depth and detail. Secondly, the interviews were conducted promptly, within 24 hours of the doctor's discharge instructions. This approach helped mitigate the potential recall bias that could arise over time. Furthermore, it is reasonable to assume that patients' responses, obtained when they faced the immediate prospect of discharge, provided a more real-time and emotionally responsive perspective on the issues related to discharge preparation care delivery.

## Conclusion

Findings suggest that patients undergoing fast-track TKA report good discharge preparation experiences. However, closer analysis reveals difficulties with this process, highlighted by psychological activities such as feelings of helplessness, stigma, and anxiety associated with uncomfortable symptoms. Pain control, attention to environmental modifications and strengthening of social and family support are important to improve postoperative management. Finally, discharge preparation nursing services can continue to work on a cordial bedside manner, listening to the patient, and humanizing the continuum of care or referral services.

## Supporting information

**S1 Checklist. Consolidated criteria for reporting qualitative research (COREQ) checklist.**
(PDF)

**S1 File. Details of the fast-track clinical components.**
(DOCX)

**S2 File. Discharge information for patients.**
(DOCX)

**S3 File. Patient interview schedule.**
(DOCX)

**S1 Table. The structural analysis and how the themes and subthemes emerged.**
(DOCX)

## Acknowledgments

The authors thank all the patients giving their time to participate in this study.

## Author Contributions

**Conceptualization:** Hong Ji.

**Data curation:** Simeng You, Na Li, Hong Ji.

**Formal analysis:** Simeng You, Na Li.

**Funding acquisition:** Hong Ji.

**Investigation:** Simeng You.

**Methodology:** Na Li, Manjie Guo.

**Resources:** Simeng You, Hong Ji.

**Supervision:** Manjie Guo, Hong Ji.

**Validation:** Manjie Guo, Hong Ji.

**Writing – original draft:** Simeng You.

**Writing – review & editing:** Simeng You.

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
