## [Decision Letter · Decision Letter 0]

27 Dec 2023

PONE-D-23-33287Are patients ready for discharge from the hospital after fast-track total knee arthroplasty? -A qualitative studyPLOS ONE

Dear Dr. Ji,

Thank you for submitting your manuscript to PLOS ONE. After careful consideration, we feel that it has merit but does not fully meet PLOS ONE’s publication criteria as it currently stands. Therefore, we invite you to submit a revised version of the manuscript that addresses the points raised during the review process.

Please revise. 

We look forward to receiving your revised manuscript.

Kind regards,

Academic Editor

PLOS ONE

“This study is funded by the Medical and Health Science and Technology Development Program of Shandong Province, China (202014021227).”

Reviewers' comments:

Reviewer's Responses to Questions

**Comments to the Author**

1. Is the manuscript technically sound, and do the data support the conclusions?

Reviewer #1: Yes

Reviewer #2: No

2. Has the statistical analysis been performed appropriately and rigorously? 

Reviewer #1: N/A

Reviewer #2: No

3. Have the authors made all data underlying the findings in their manuscript fully available?

Reviewer #1: Yes

Reviewer #2: Yes

4. Is the manuscript presented in an intelligible fashion and written in standard English?

Reviewer #1: Yes

Reviewer #2: Yes

5. Review Comments to the Author

Reviewer #1: It is not clear what this study adds to existing knowledge. Please better justify the present study taking into account existing literature.

https://pubmed.ncbi.nlm.nih.gov/26708610/

https://pubmed.ncbi.nlm.nih.gov/31769559/

https://pubmed.ncbi.nlm.nih.gov/30297138/

https://pubmed.ncbi.nlm.nih.gov/31227003/

https://pubmed.ncbi.nlm.nih.gov/27345939/

Line 86: what is the evidence of the benefits of patient involvement?

Lines 95 to 100: the objective needs to be made clearer.

Line 98: this goes beyond nursing and includes physicians and rehab specialists.

Line 124:

-More details are needed on TKA patient characteristics: due to OA only or other health problems such as fractures, RA, primary elective, etc? Were revisions and bilaterals included or not?

-Among these patients, describe how are patients selected for fast-track

-Details of the fast-track clinical components are needed. It is not clear what this program consists of.

Line 138: who provided the training?

Line 142: provide in detail what information is given to patients before and after surgery regarding discharge, both paper and orally (could be given in an appendix).

-Line 152: on how many patients?

-Line 153: describe in detail how the guide was developed.

-Line 160; How was this taken into account in the analyses?

-Table 1: report medians and not means. What does postoperative period mean? Report hospital length of stay data. If I guess LOS correctly, it seems quite high for a fast-track program. How was the number of chronic diseases determined?

-Line 204: submitting to medical authority seems a bit harsh. What about "preparing for discharge despite..."

-Line 256: replace gaming with managing.

-Line 281: their

-Line 386: do patients pay for hospital stay? Please add this context in the manuscript

-Line 409: The link with the results needs to be made clearer. Can you be more explicit on what subjects?

Line 419 and 422: please replace paranoid.

-Line 421: probably not all patients.

-Line 460: please replace infected.

-Line 470: the results are limited to the Chinese health care system.

Reviewer #2: Abstract: Very vaguely presented.

Relevant details need to be added

Intro: Very long. The purpose of the study is not clear

Relevant literature may be appropriately presented

Methods: "We purposively recruited patients": Does not read well> Pls modify the inclusion criteria

Results: Does not follow a good scientific flow.

Predominantly psychological issues have been discussed. No other clinical information of patients has been shared

Discussion: Very superficial. The manuscript does not have a scientific flow.

The purpose of the study is not clarified

Inadequate details have been furnished

Conclusion: Long and not very clear.

6. PLOS authors have the option to publish the peer review history of their article (what does this mean?). If published, this will include your full peer review and any attached files.

Reviewer #1: No

Reviewer #2: No

---

## [Author Response · Author response to Decision Letter 0]

9 Feb 2024

Editor comments: Thank you for submitting your manuscript to PLOS ONE. After careful consideration, we feel that it has merit but does not fully meet PLOS ONE’s publication criteria as it currently stands. Therefore, we invite you to submit a revised version of the manuscript that addresses the points raised during the review process.

Author response: Thank you for the opportunity to submit a revised version of the manuscript. We greatly appreciate the reviewers’ comments and consider addressing these have led to an improved manuscript. We can confirm the manuscript meets PLOS ONE's style requirements, including those for file naming. Please find our response to the reviewers’ comments below. We have also included the manuscript changes and used red type below to highlight the relevant changes to the manuscript text.

Reviewer #1: It is not clear what this study adds to existing knowledge. Please better justify the present study taking into account existing literature.

Author response: Thank you for your question and for providing us with articles that need to be re-read carefully for this purpose. Our response is as follows: First, a review of the literature found that current qualitative researches on this subject have focused primarily on the staged experiences of patients undergoing fast-track total knee/hip arthroplasty (TKA/THA) throughout the discharge process or in the short-term post-discharge period, and appeared to have limited understanding of patients' experiences at the time of receiving discharge instructions. The time point at this study was chosen to take place within 24 hours of receiving discharge instructions anticipating more feedback from patients as they face the decision to be discharged. Second, we recognized that this study did share similarities with the results of prior studies, such as poor pain management, uncertainty and lack of confidence in recovery due to poor information, stigma, and social relationships still exist. However, in contrast to previous studies, this study, in addition to revealing the challenges and shortcomings faced by patients, was fortunate to hear some positive responses from patients regarding clinical care practices, such as compassionate bedside manner, listening to the patient's voice, and humanized continuity of care and referral services. This also has implications for the continued maintenance and renewed efforts to improve discharge care services in the future.

Reviewer #1: Line 86: What is the evidence of the benefits of patient involvement?

Author response: Thank you very much for your comments, and after reviewing the relevant literatures, we added the benefits of patient involvement in the revised manuscript, including reduced unplanned readmission and mortality rates, improved patient satisfaction, and also helping to optimize the discharge planning process. References have also been added to the revised manuscript.

Reviewer #1: Lines 95 to 100: the objective needs to be made clearer.

Author response: Thank you very much for your suggestion, and in the revised manuscript we have clarified the objectives of this study even more.

Actually, this study contains two explicit objectives. Objective 1: To describe the discharge preparation experiences of patients after fast-track TKA and to clarify whether patients are consciously prepared for discharge. Objective 2: To explore what gaps exist in the current discharge preparation care service for TKA under fast-track and what can be improved.

Reviewer #1: Line 98: this goes beyond nursing and includes physicians and rehab specialists.

Author response: Many thanks for pointing this out. This does include not only the scope of nursing, but also physicians and rehabilitation specialists, and given the readership of the results of this study, we decided to remove this item to ensure that it falls within the scope of nursing responsibilities.

Reviewer #1: Line 124: -More details are needed on TKA patient characteristics: due to OA only or other health problems such as fractures, RA, primary elective, etc? Were revisions and bilaterals included or not?

Author response: We added more details on patient characteristics. In this study, we only considered patients who underwent their first unilateral TKA for advanced osteoarthritis of the knee (KOA), excluding fractures, rheumatoid arthritis, or revisions. In addition, all patients we included for interview were primary elective.

Reviewer #1: Line 124: -Among these patients, describe how are patients selected for fast-track

Author response: Thank you very much for your suggestion. Patients recruitment for this study was from May 2023 to July 2023. From May 2023, all elective patients (ASA classification I, II and stable III) diagnosed with osteoarthritis of the knee and scheduled to undergo TKA were enrolled in the fast-track system. We have added this to the text as requested.

Reviewer #1: Line 124: -Details of the fast-track clinical components are needed. It is not clear what this program consists of.

Author response: Thanks for your kind suggestions, which are valuable for improving the accuracy of the manuscript. In the revised manuscript, we have added details about the clinical components of the fast-track, which we have placed in the S1 File (Details of the fast-track clinical components) for your review.

Reviewer #1: Line 138: who provided the training?

Author response: The theory of qualitative research was studied and assessed in the School of Nursing, and the internship hospital (Research Department of the First Affiliated Hospital of Shandong First Medical University) provided training on interviewing techniques.

Reviewer #1: Line 142: provide in detail what information is given to patients before and after surgery regarding discharge, both paper and orally (could be given in an appendix).

Author response: Thank you for your advice. Discharge information mainly includes: informing of discharge criteria, functional exercises, proper diet, medication instructions, appropriate home set-up and competent adult caregivers, and instructions for daily living. We have included this section in S2 File (Discharge Information for Patients) for your reference.

Reviewer #1: Line 152: on how many patients?

Author response: This pilot study included a total of 3 patients.

Reviewer #1: Line 153: describe in detail how the guide was developed.

Author response: Thank you for your advice, and we have added specific details in the revised manuscript, which you are invited to review.

Reviewer #1: Line 160: How was this taken into account in the analyses?

Author response: Thank you very much for asking this question, and our response is as follows:The interview questions involved probing patients' true feelings about their readiness for discharge, as well as patients' comments and suggestions about the quality of care they received on discharge, and although we explained to patients that we were not responsible for physical health assessments and quality of care checks within the unit, participants may have still reported what they thought the researchers would want to hear (this is a point we have already touched upon in the limitations of this study), and we preferred to hear patients' most honest expression of their views. Therefore, in addition to verbal evidence, non-verbal evidence (voice tones, expressions, body movements, etc.) was considered, such as facial expressions to get an idea of the level of joint pain and the veracity of the patient's assessment of the care team; tone of voice to determine the patient's level of confidence in being discharged from the hospital; and flexion/extension of the knee to determine the patient's current joint function. This really helped the researchers to better summarize the themes.

Reviewer #1: Table 1: report medians and not means. What does postoperative period mean? Report hospital length of stay data. If I guess LOS correctly, it seems quite high for a fast-track program. How was the number of chronic diseases determined?

Author response: Thank you for your valuable comments. First, we replaced the means with the medians and the postoperative period with the hospital length of stay(LOS) in the table 1. Secondly, our answer to the question that the LOS data seems quite high is as follows: in reviewing previous studies, the LOS for TKA under fast track has been reduced in the last decade in the USA[1], Canada[2], and the UK[3], etc., to about 2-3 days postoperatively, but the average national LOS is still 3-5 days[4, 5]. The median LOS in this study was 3 days, which is an acceptable result. For the individual patient in this study who had LOS as long as 5 days, we can find that the patient was characterized by female, older than 80 years of age, and living with caregivers only, etc. For the patient, physicians consider moderately lengthening the LOS in order to prevent postoperative complications such as joint pain, inflammatory response, upright intolerance, and cognitive dysfunction, and to allow some of these high-risk patients to benefit from a planned extension of their hospital stays. Finally, regarding the determination of the number of chronic diseases, this study was obtained by reviewing patient case files and through verbal re-interviews. We pre-listed a number of common and highly prevalent chronic diseases, including hypertension, diabetes mellitus, arrhythmia, atherosclerosis, coronary heart disease, osteoporosis, eye disease, hearing impairment, stroke, obesity, cancer, gastritis, gout, bronchial asthma, and so on, and set up the option of "other" for patients to add.

Reviewer #1: Line 204: submitting to medical authority seems a bit harsh. What about "preparing for discharge despite..."

Author response: Thank you for suggesting changes to the theme, after our consideration and communication, we unanimously agree with your suggestions and have now made the changes.

Reviewer #1: Line 256: replace gaming with managing.

Author response: Thank you for suggesting changes to the refinement of the theme of this study, which has now been revised.

Reviewer #1: Line 281: their

Author response: Thank you very much for pointing out the spelling errors in the words that appeared in our article, we apologize for this and have now made corrections and re-reviewed to ensure that there are no spelling problems with the words in this article.

Reviewer #1: Line 386: do patients pay for hospital stay? Please add this context in the manuscript

Author response: Yes. Considering that the occupation of the bed, nursing services and treatment, patients are required to pay for the hospital stay. We will add the relevant context in the revised manuscript.

Reviewer #1: Line 409: The link with the results needs to be made clearer. Can you be more explicit on what subjects?

Author response: Thank you for your suggestion, we think it helps to explain the subjects of this study more clearly and we have rewritten this section based on the reviewers' comments. We offer this recommendation based on the findings of Theme 1(Preparing for discharge despite concerns about symptoms), which included the sub-themes of satisfaction with improved function and a sense of reborn joy, feelings of helplessness, stigmatization, and anxiety about prosthetic function. (Sub-themes included in theme are presented in the results of the revised manuscript.)

Reviewer #1: Line 419 and 422: please replace paranoid.

Author response: We appreciate your professional comments about our inaccurate terminology, and based on your suggestions, we have replaced it with “stubborn” in previous manuscript.

Reviewer #1: Line 421: probably not all patients.

Author response: Thank you for your question regarding the scope of the population involved. After further discussion, we realized that this was a misunderstanding, and for the sake of clarity, we will refer to "patients" as "some patients".

Reviewer #1: Line 460: please replace infected.

Author response: Thank you for your suggestion about the poor use of words in this study, which we correct immediately, and we have decided to replace the word "infected" with the word "impacted".

Reviewer #1: Line 470: the results are limited to the Chinese health care system.

Author response: Thank you very much for revising the limitations of this study, we have changed the limitation of this study to the fact that the results are limited to the Chinese health care system.

Reviewer #2: Abstract: Very vaguely presented. Relevant details need to be added

Author response: We sincerely thank you for your valuable suggestions and have added details in each section of the abstract such as background, objectives, methods, results and conclusion respectively.

Reviewer #2: Intro: Very long. The purpose of the study is not clear. Relevant literature may be appropriately presented

Author response: Thank you for your valuable comments, we have re-read the Introduction and in the revised version, we have deleted repetitively expressed sentences to streamline the content. Also, we have made the purpose of this study more explicit, which contains two objectives.

Objective 1: To describe the discharge preparation experiences of patients after fast-track TKA and to clarify whether patients are consciously prepared for discharge. Objective 2: To explore what deficiencies exist in the current discharge preparation nursing service for TKA under fast-track and what can be improved. Moreover, we scrutinized the literature and added more references in the introduction section of the revised manuscript.

Reviewer #2: Methods: "We purposively recruited patients": Does not read well. Pls modify the inclusion criteria

Author response: We found your suggestions very helpful and we have added more details on patient characteristics and modified the inclusion criteria. We have detailed that patients were first elective unilateral TKA due to advanced KOA, excluding fractures, rheumatoid arthritis, or revisions. In addition, all patients we included for interview were primary elective procedures.

Reviewer #2: Results: Does not follow a good scientific flow. Predominantly psychological issues have been discussed. No other clinical information of patients has been shared

Author response: Thanks for your suggestions, we have rewritten this section based on the reviewer's comments. During the process, we did our best to follow a good scientific flow to report only results-related content and to avoid excessive discussion in this section. Moreover, in order to present the results more clearly, sub-themes have been added to aid understanding (this section was shown further in S1 Table). Finally, in addition to emphasizing psychological issues, other details were included, which we have reflected in the revised manuscript.

Reviewer #2: Discussion: Very superficial. The manuscript does not have a scientific flow. The purpose of the study is not clarified. Inadequate details have been furnished

Author response: We sincerely thank you for your careful reading, and we have fully considered your suggestions and made our best efforts to revise the manuscript, rewrite it according to the scientific process, clarify the purpose of this study and provide relevant details.

Reviewer #2: Conclusion: Long and not very clear.

Author response: We sincerely thank you for pointing out our shortcomings, and we have revisited the relevant elements of our conclusions and have streamlined and clarified the conclusions of this study in the revised manuscript.

---

## [Decision Letter · Decision Letter 1]

22 Feb 2024

PONE-D-23-33287R1Are patients ready for discharge from the hospital after fast-track total knee arthroplasty? -A qualitative studyPLOS ONE

Dear Dr. Ji,

Thank you for submitting your manuscript to PLOS ONE. After careful consideration, we feel that it has merit but does not fully meet PLOS ONE’s publication criteria as it currently stands. Therefore, we invite you to submit a revised version of the manuscript that addresses the points raised during the review process.

Please revise.

We look forward to receiving your revised manuscript.

Kind regards,

Academic Editor

PLOS ONE

Reviewers' comments:

Reviewer's Responses to Questions

**Comments to the Author**

1. If the authors have adequately addressed your comments raised in a previous round of review and you feel that this manuscript is now acceptable for publication, you may indicate that here to bypass the “Comments to the Author” section, enter your conflict of interest statement in the “Confidential to Editor” section, and submit your "Accept" recommendation.

Reviewer #1: (No Response)

Reviewer #2: (No Response)

2. Is the manuscript technically sound, and do the data support the conclusions?

Reviewer #1: Yes

Reviewer #2: No

3. Has the statistical analysis been performed appropriately and rigorously? 

Reviewer #1: Yes

Reviewer #2: No

4. Have the authors made all data underlying the findings in their manuscript fully available?

Reviewer #1: Yes

Reviewer #2: No

5. Is the manuscript presented in an intelligible fashion and written in standard English?

Reviewer #1: Yes

Reviewer #2: Yes

6. Review Comments to the Author

Reviewer #1: Line 307: replace medicines with medication

L:ine 480 and 483: I don't think stubborn is the right word either in the context of patient-centered care. The patient's feelings should be more respectfully acknowledged, and a term more patient-centered should be used. I would suggest rephrasing to simply use the patient's resistance to use analgesics, and remove any paternalistic terms.

Reviewer #2: We appreciate the authors hard work

The authors goal is to evaluate whether the patients are discharge ready for fast-track discharges

However, the manuscript does not really serve the goal.

The manuscript is less scientific; and more anecdotal in its approach

7. PLOS authors have the option to publish the peer review history of their article (what does this mean?). If published, this will include your full peer review and any attached files.

Reviewer #1: No

Reviewer #2: No

---

## [Author Response · Author response to Decision Letter 1]

24 Mar 2024

Additional editor comments

Thank you for submitting your manuscript to PLOS ONE. After careful consideration, we feel that it has merit but does not fully meet PLOS ONE’s publication criteria as it currently stands. Therefore, we invite you to submit a revised version of the manuscript that addresses the points raised during the review process.

Author response: Thank you for the opportunity to submit a revised version of the manuscript. We greatly appreciate the reviewers’ comments and consider addressing these have led to an improved manuscript. We can confirm the manuscript meets PLOS ONE's style requirements, including those for file naming. Please find our response to the reviewers’ comments below. We have also included the manuscript changes and used red type below to highlight the relevant changes to the manuscript text.

Reviewer #1: Line 272: replace medicines with medication

Author response: Thanks for your kind suggestions, which are valuable for improving the accuracy of the manuscript. We have made a revision to this.

Reviewer #1: Line 480 and 483: I don't think stubborn is the right word either in the context of patient-centered care. The patient's feelings should be more respectfully acknowledged, and a term more patient-centered should be used. I would suggest rephrasing to simply use the patient's resistance to use analgesics, and remove any paternalistic terms.

Author response: Thank you very much for your suggestions, and in the revised manuscript we have paid further attention to the use of word and removed some of the paternalistic terms

Reviewer #2: We appreciate the authors hard work. The authors goal is to evaluate whether the patients are discharge ready for fast-track discharges. However, the manuscript does not really serve the goal. The manuscript is less scientific; and more anecdotal in its approach

Author response: Thank you very much for reading this paper carefully and for making meaningful recommendations. In response to the goal of this study, the primary aim of this study was 1) to increase the knowledge about patients’ experiences of discharged from hospital via a fast-track process after TKA. The secondary aim was 2) to explore what gaps exist in the current discharge preparation care service for TKA under fast-track and what can be improved.In terms of evaluating whether the patients are ready for fast-track discharges, this study was a qualitative study in nature and could not really achieve this goal, which is related to the methodology that distinguishes qualitative studies from quantitative studies. This study was only able to discern that patients expressed a good experience of discharge readiness, and relevant inappropriate statements in the text were modified or removed in this manuscript.

Regarding the less of scientific in this manuscript, and more anecdotal in its approach. After receiving comments from the reviewer, the joint research team revisited and understood the thick descriptive content and revised the presentation of the data excerpts provided in the results.

---

## [Decision Letter · Decision Letter 2]

3 May 2024

Are patients ready for discharge from the hospital after fast-track total knee arthroplasty? -A qualitative study

PONE-D-23-33287R2

Dear Dr. Ji,

We’re pleased to inform you that your manuscript has been judged scientifically suitable for publication and will be formally accepted for publication once it meets all outstanding technical requirements.

Kind regards,

Academic Editor

PLOS ONE

Additional Editor Comments (optional):

Reviewers' comments:

Reviewer's Responses to Questions

**Comments to the Author**

1. If the authors have adequately addressed your comments raised in a previous round of review and you feel that this manuscript is now acceptable for publication, you may indicate that here to bypass the “Comments to the Author” section, enter your conflict of interest statement in the “Confidential to Editor” section, and submit your "Accept" recommendation.

Reviewer #1: All comments have been addressed

Reviewer #3: All comments have been addressed

2. Is the manuscript technically sound, and do the data support the conclusions?

Reviewer #1: Yes

Reviewer #3: Yes

3. Has the statistical analysis been performed appropriately and rigorously? 

Reviewer #1: Yes

Reviewer #3: I Don't Know

4. Have the authors made all data underlying the findings in their manuscript fully available?

Reviewer #1: Yes

Reviewer #3: Yes

5. Is the manuscript presented in an intelligible fashion and written in standard English?

Reviewer #1: Yes

Reviewer #3: Yes

6. Review Comments to the Author

Reviewer #1: (No Response)

Reviewer #3: (No Response)

7. PLOS authors have the option to publish the peer review history of their article (what does this mean?). If published, this will include your full peer review and any attached files.

Reviewer #1: No

Reviewer #3: No

---

## [Editor Report · Acceptance letter]

16 May 2024

PONE-D-23-33287R2 

PLOS ONE

Dear Dr. Ji, 

I'm pleased to inform you that your manuscript has been deemed suitable for publication in PLOS ONE. Congratulations! Your manuscript is now being handed over to our production team.

Kind regards, 

on behalf of

Dr. Robert Jeenchen Chen 

Academic Editor

PLOS ONE